# FedEAT: A Robustness Optimization Framework for Federated LLMs

## Abstract

The integration of federated learning (FL) with large language models (LLMs) leverages the privacy-preserving benefits of decentralized data processing in sensitive domains such as healthcare, finance, and law, while also addressing the growing scarcity of high-quality training data for LLMs. However, in practical deployments, federated large language models (federated LLMs) are highly vulnerable to adversarial attacks, which can severely undermine their reliability and stability. To overcome these challenges, we introduce fedEAT (Federated Embedding-Space Adversarial Training), a novel algorithm that performs adversarial training directly in the client LLM's embedding space and incorporates a regularization term to balance robustness against clean-data accuracy. Extensive experiments demonstrate that, compared to conventional federated LLMs, fedEAT greatly enhances classification accuracy on adversarial examples, while causing only negligible performance degradation on clean inputs, and remains scalable to tasks in other domains. These results validate fedEAT's effectiveness and practical value in enhancing the robustness of federated LLMs across critical, privacy-sensitive applications.

## 1 Introduction

Large Language Models (LLMs) have achieved breakthrough advances in tasks such as natural language understanding, code generation, and text creation Nam et al. (2024); Tang et al. (2024); Yao et al. (2024). However, their reliance on substantial compute resources and large volumes of high-quality data for pre-training and fine-tuning makes it difficult for any single organization to support them independently. At the same time, regulatory and privacy constraints often lead to data silos across organizations, preventing effective sharing of core assets Bai et al. (2024); Ye et al. (2024); Yue et al. (2024b).

Federated Learning (FL) addresses these issues by keeping raw data local and sharing only model updates, thereby breaking down data silos and unifying distributed compute resources—an effective solution for privacy-sensitive domains Mammen (2021); Yue et al. (2024a).

Table 1: Comparison of adversarial robustness definitions.

| Setting | Related work | Robustness definition |
|---|---|---|
| Federated learning | Dong & Xu (2023); Zhou et al. (2022); Zizzo et al. (2020) | Maintain correct classification performance under adversarial examples. |
| LLMs | Xhonneux et al. (2024); Mazeika et al. (2024); Schwinn et al. (2023) | Resist adversarial prompt attacks and refuse to generate harmful content. |
| Federated LLMs | Ours | Maintain task performance during inference when facing untargeted adversarial examples. |

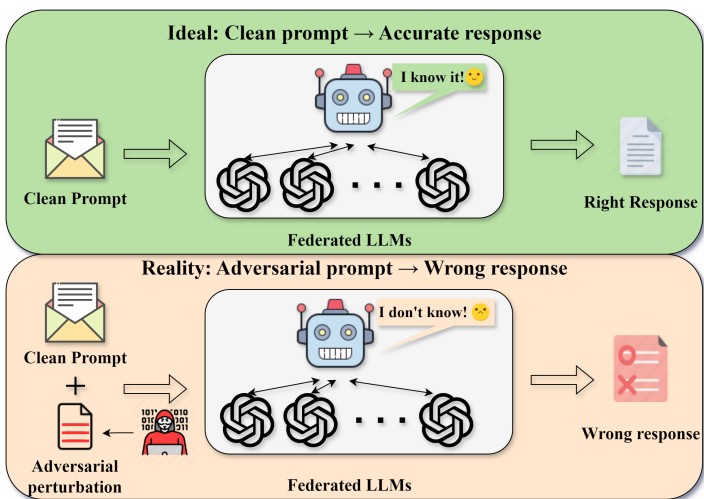

Figure 1: adversarial robustness challenge in federated LLMs. during inference, the model is vulnerable to malicious or random perturbations that can degrade its performance.

Adversarial samples—inputs with carefully crafted perturbations—pose a critical challenge to both traditional FL and LLMs, as real-world inputs are often noisy, and adversarial noise represents the most likely type of perturbation to cause model failures. Small changes in the input can drastically reduce classification accuracy or lead to harmful outputs. As shown in Table 1, FL, commonly applied to image classification, focuses on "maintaining classification performance under adversarial samples" Dong & Xu (2023); Zhou et al. (2022); Zizzo et al. (2020), whereas cloud-deployed LLMs emphasize "resisting adversarial prompts and refusing to generate harmful content." By improving adversarial robustness, models can also become more resilient to general input noise.Xhonneux et al. (2024); Mazeika et al. (2024); Schwinn et al. (2023). When FL is combined with LLMs, the resulting federated LLMs unlock significant potential in privacy-sensitive domains such as healthcare, finance, and lawSani et al. (2024). However, in these high-risk applications, subtle term modifications in medical records can lead to misdiagnoses, slight perturbations in financial comments can mislead risk systems, and clause tweaks in legal documents can cause retrieval and archiving errors. To address these challenges, we define "adversarial robustness" for federated LLMs as the ability to maintain original task performance during inference when facing untargeted adversarial examples.

While adversarial training is a time-tested approach for adversarial robustness and has shown notable success in FL settingsZizzo et al. (2020), extending it to federated LLMs with billions to trillions of parameters introduces severe hurdles. First, the extreme gradient dimensionality and sheer parameter count drive computation and memory costs to explode, making conventional multi-step PGD methods untenable. Second, generating adversarial examples in the discrete text domain is highly time-consuming, rendering existing schemes impractical in resource-constrained federated environments.

Accordingly, federated LLMs must address two key challenges that go beyond those encountered in traditional FL and centralized LLMs: compute and memory efficiency, and distributed robustness with communication bottlenecks. Traditional FL can efficiently produce adversarial samples via multi-step PGD, and federated BERTs (hundreds of millions of parameters) can run FGSM or PGD on-device. However, for billion- to trillion-parameter LLMs, the vast gradient and embedding spaces make multi-step adversarial training prohibitively expensive in both computation and memory. Centralized LLMs can leverage full access to the global data distribution for robust adversarial training, but federated LLMs only aggregate updates from clients' private data, limiting their ability to capture global adversarial patterns under non-IID distributions—resulting in uneven robustness across client domains. Moreover, full-parameter aggregation in a federated setup incurs massive communication overhead, whereas centralized training faces no such bandwidth constraints.

To address these challenges, we propose fedEAT (federated Embedding-Space Adversarial Training), which (1) generates directed perturbations in the client LLM's embedding space to significantly

reduce the computational overhead of adversarial sample construction; (2) introduces a regularization term in the loss function to balance adversarial robustness and clean-input performance; and (3) integrates the LoRA training strategy to substantially decrease communication overhead during federated LLMs training.

Our main contributions are as follows:

- We systematically analyze the adversarial robustness challenges faced by federated LLMs at inference time and formulate the corresponding optimization objectives.

- We design the fedEAT algorithm based on embedding-space adversarial training and introduce a regularization term to balance robustness and accuracy.

- We conduct extensive large-scale empirical studies in high-privacy domains such as healthcare, finance and law, as well as across various text classification tasks, verify that fedEAT delivers significant improvements in adversarial robustness and exhibits strong scalability to other domains.

## 2   RELATED WORKS

In this section, we will discuss the robustness challenges associated with federated learning(FL) and large language models(LLMs). Then, we will review research on federated LLMs. The comparison table of related work is provided in the appendix Table 7.

### 2.1   ADVERSARIAL ROBUSTNESS

In federated learning, adversarial attacks exploit subtle input perturbations—adversarial examples—to degrade model accuracy under attack Bhagoji et al. (2019). Defenses such as robust aggregation Pillutla et al. (2022), adversarial training Zizzo et al. (2020), and anomaly detection Mothukuri et al. (2021) aim to preserve correct classification performance, but practical robustness remains elusive.

For large language models, studies have focused on "robust refusal" in text generation via adversarial training on harmful-prompt datasets Xhonneux et al. (2024); Mazeika et al. (2024), though no standard benchmark exists Schwinn et al. (2023). These methods do not generalize to downstream tasks like text classification, where robustness means maintaining accuracy against input perturbations rather than rejecting prompts. More efficient, scalable algorithms are urgently needed to secure LLMs across diverse applications.

### 2.2   FEDERATED LARGE LANGUAGE MODELS

Federated LLMs marry federated learning (FL) with large language models (LLMs) to enable collaborative, privacy-preserving training across distributed clients Sani et al. (2024). As illustrated in Figure 2, the server and clients share an identical LLM architecture, but clients—constrained by compute—fine-tune only a low-rank subset of parameters via LoRA. Prior work has also explored hybrid setups where clients use smaller language models and receive distilled knowledge from a server LLM Fan et al. (2024a;b), yet for simplicity we assume uniform LLMs throughout. Training proceeds in FL's usual round-based fashion: the server broadcasts parameters, clients update on local data, and the server averages the returned updates. Despite this promise, federated LLMs introduce unique robustness challenges—such as large-scale adversarial attacks and communication-constrained defenses—that remain largely unexplored.

## 3   METHOD

In this section, we introduce fedEAT, our federated embedding-space adversarial training algorithm. We first define the problem we aim to solve, then describe the process of generating adversarial examples in the embedding space of LLMs, and finally provide a detailed exposition of the fedEAT algorithm.

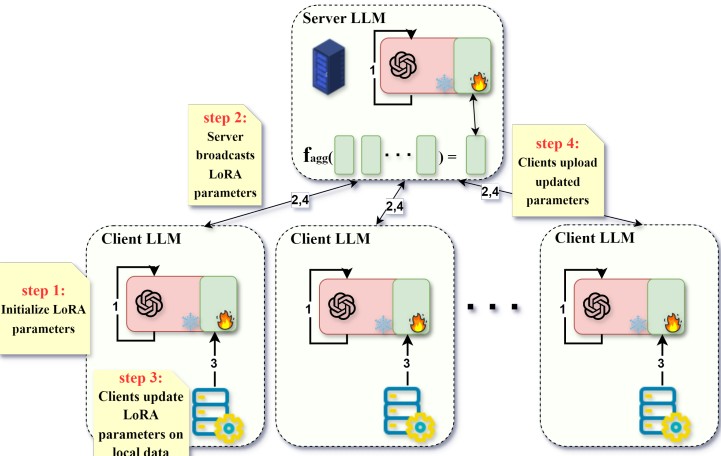

Figure 2: This figure illustrates the training framework for federated LLMs. As the computational constraints of client devices, client LLMs typically adopt Low-Rank Adaptation (LoRA) to fine-tune only a designated subset of the model parameters.

### 3.1 PROBLEM DEFINITION

In the federated LLMs framework, $K$ clients collaboratively train a global LLM without sharing their raw data. We focus on the robustness of such models under adversarial attacks, aiming to enhance overall adversarial robustness without compromising the model's original task performance.

To achieve this, we first formally define the adversarial robustness optimization problem for federated LLMs. Let $\mathcal{D}_k$ represent the local dataset of client $k$ with size $|\mathcal{D}_k|$, and $\theta$ denote the global model parameters. The original loss is defined as:

$$\mathcal{L}_{\text{orig}}(\theta) = \sum_{k=1}^{K} \frac{1}{|\mathcal{D}_k|} \sum_{(x,y) \in \mathcal{D}_k} \ell\big(f_\theta(x),\, y\big), \tag{1}$$

where $\ell(f_\theta(x), y)$ represents the task-specific loss for sample $(x, y)$.

To evaluate model performance under adversarial perturbations, we define the adversarial loss:

$$\mathcal{L}_{\text{adv}}(\theta) = \sum_{k=1}^{K} \frac{1}{|\mathcal{D}_k|} \sum_{(x,y) \in \mathcal{D}_k} \max_{\|\delta\| \leq \epsilon} \ell\big(f_\theta(x + \delta),\, y\big), \tag{2}$$

where $\delta$ represents an adversarial perturbation constrained by $\|\delta\| \leq \epsilon$ to limit semantic deviation.

Our final optimization objective balances between the original and adversarial losses:

$$\min_{\theta} (1 - \lambda)\, \mathcal{L}_{\text{orig}}(\theta) \,+\, \lambda\, \mathcal{L}_{\text{adv}}(\theta), \quad \lambda \in [0, 1], \tag{3}$$

where the hyperparameter $\lambda$ controls the trade-off between model performance and robustness, enabling us to enhance overall system robustness without degrading the original task performance.

### 3.2 EMBEDDING SPACE ADVERSARIAL SAMPLE GENERATION

Prior work has demonstrated the effectiveness of generating adversarial examples in continuous spaces Xhonneux et al. (2024); Schwinn et al. (2023). To avoid the high cost of crafting adversarial samples in the discrete text domain, we instead generate perturbations directly in the embedding space of federated LLMs. As illustrated in Figure 3, we first tokenize the input and map it to an embedding vector $\mathbf{z} = E(\mathbf{x})$; then, on each client LLM, we iteratively update the perturbation vector along the gradient direction according to

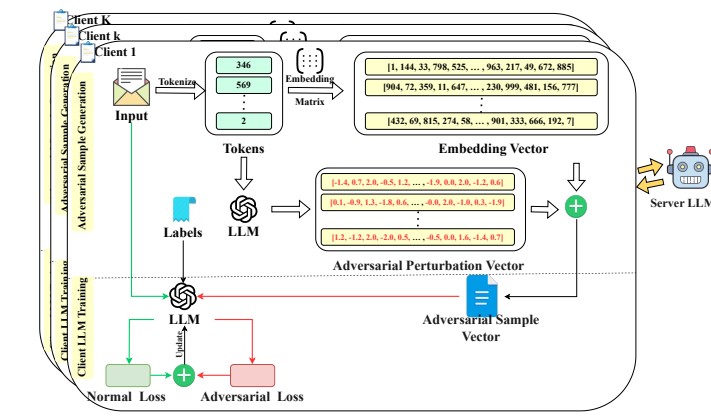

Figure 3: This figure illustrates the generation of adversarial samples in the embedding space and the fedEAT algorithm.

$$\delta^{t+1} = \Pi_{\mathcal{B}(0,\epsilon)}\Big(\delta^t + \alpha \operatorname{sign}\big(\nabla_{\mathbf{z}}\mathcal{L}(f_\theta(\mathbf{z}+\delta^t), y))\big)\Big), \quad t = 0, \ldots, T-1, \tag{4}$$

where $\alpha$ is the step size, and $\Pi_{\mathcal{B}(0,\epsilon)}$ projects the updated vector back into the $L_p$-ball to enforce the constraint $\|\delta\|_p \leq \epsilon$. After $T$ iterations, we obtain the final adversarial embedding

$$\mathbf{z}_{\mathrm{adv}} = \mathbf{z} + \delta^T. \tag{5}$$

During training, these perturbations are optimized to maximize the adversarial loss, producing the most damaging examples, which are then used to reinforce the federated LLM's robustness.

### 3.3 FedEAT: A Robustness Optimization Framework

The fedEAT algorithm aims to significantly enhance the adversarial robustness of federated LLMs while preserving their original performance, addressing the optimization problem presented in Section 3.1.

First, fedEAT relocates adversarial sample generation from discrete text space to continuous embedding space: clients only need to map inputs to embedding vectors $\mathbf{z} = E(\mathbf{x})$, then iteratively generate adversarial perturbations $\delta$ along gradient directions to construct adversarial samples. This approach substantially reduces computational overhead and optimization difficulty, making it particularly suitable for resource-constrained edge devices.

Second, fedEAT incorporates joint training with original and adversarial samples: during each round of federated communication, the client's local optimization objective is

$$\min_\theta (1-\lambda)\, \mathcal{L}_{\mathrm{orig}}(\theta) \;+\; \lambda\, \mathcal{L}_{\mathrm{adv}}(\theta), \quad \lambda \in [0,1], \tag{6}$$

which uses the hyperparameter $\lambda$ to balance model performance on clean inputs and adversarial inputs, enabling it to both resist attacks and maintain performance on downstream tasks such as classification or generation.

Finally, to address the communication bottleneck in models with hundreds of billions of parameters, fedEAT adopts the LoRA low-rank fine-tuning strategy.

As illustrated in Figure 3, FedEAT proceeds in communication rounds where the server first broadcasts the current global LoRA parameters $\mathbf{w}^{(t)}$ to all clients, which initialize their local models accordingly. Each client then constructs adversarial examples by embedding each input $x$ into $\mathbf{z} = \operatorname{Embed}(x)$, initializing $\delta^0 = \mathbf{0}$ and performing $I$ steps of gradient-based updates in the embedding space, projecting onto the $\epsilon$-norm ball at each step to obtain $\mathbf{z}_{\mathrm{adv}}$. The client computes a combined loss

$$L = (1-\lambda)\,\mathcal{L}(f_\theta(\mathbf{z}), y) + \lambda\,\mathcal{L}(f_\theta(\mathbf{z}_{\mathrm{adv}}), y),$$

and updates only its local LoRA parameters $\theta_k$ via gradient descent. Finally, all clients upload their updated $\theta_k$ to the server, which aggregates them by averaging to form the new global parameters $\mathbf{w}^{(t+1)} = \frac{1}{n} \sum_{k=1}^{n} \theta_k$. Algorithm 1 provides a detailed specification of this process.

---

**Algorithm 1** FedEAT: Federated Embedding-space Adversarial Training

---

**Require:** Global model $\mathbf{w}^{(0)}$, local datasets $\{D_k\}_{k=1}^{n}$, communication rounds $T$, local epochs $E$, learning rate $\eta$, perturbation step $\alpha$, norm bound $\epsilon$, trade-off coefficient $\lambda$, iteration rounds $I$
**Ensure:** Global model parameters $\mathbf{w}^{(T)}$
1: **for** $t = 0$ to $T - 1$ **do**
2:     Server broadcasts $\mathbf{w}^{(t)}$ to all clients
3:     **for** each client $k = 1, \dots, n$ **in parallel do**
4:         $\theta_k \leftarrow \mathbf{w}^{(t)}$ {Initialize local model}
5:         **for** $e = 1$ to $E$ **do**
6:             **for** each sample $(x, y) \in D_k$ **do**
7:                 $\mathbf{z} \leftarrow \text{Embed}(x)$ {Generate embedding vector}
8:                 $\delta^0 \leftarrow \mathbf{0}$ {Initialize perturbation vector}
9:                 **for** $i = 0$ to $I - 1$ **do**
10:                     $\delta^{i+1} \leftarrow \Pi_{\mathcal{B}(0,\epsilon)}\big(\delta^i + \alpha \cdot \text{sign}(\nabla_{\mathbf{z}} \mathcal{L}(f_\theta(\mathbf{z} + \delta^i), y))\big)$
11:                 **end for**
12:                 $\mathbf{z}_{\text{adv}} \leftarrow \mathbf{z} + \delta^I$ {Generate adversarial sample}
13:                 $L \leftarrow (1 - \lambda)\,\mathcal{L}(f_\theta(\mathbf{z}), y) + \lambda\,\mathcal{L}(f_\theta(\mathbf{z}_{\text{adv}}), y)$ {Combine two losses}
14:                 $\theta_k \leftarrow \theta_k - \eta\,\nabla_{\theta_k} L$ {Update local model}
15:             **end for**
16:         **end for**
17:         Client $k$ returns $\theta_k$ to the server
18:     **end for**
19:     Server aggregation: $\mathbf{w}^{(t+1)} \leftarrow \frac{1}{n} \sum_{k=1}^{n} \theta_k$ {Simple averaging}
20: **end for**
21: **return** $\mathbf{w}^{(T)}$

---

## 4 EXPERIMENT

### 4.1 EXPERIMENTAL SETUPS

**Datasets and Model Selection:** We selected two medical text classification datasets (PubMed RCT Dernoncourt & Lee (2017) and Medical Abstracts Schopf et al. (2023)) and two legal text classification datasets (SCOTUS Yaich & Hernandez (2025) and ECHR Xu et al. (2023)) to simulate downstream tasks for federated LLMs. Additionally, following Edwards et al. Edwards & Camacho-Collados (2024), we incorporated four public text classification benchmarks (IMDB Maas et al. (2011), AG NewsZhang et al. (2016), BBC News Greene & Cunningham (2006), and Reuters Lewis (1987)) to demonstrate the cross-domain scalability of fedEAT, we partitioned the data under a non-IID setting using a Dirichlet distribution ($\alpha = 0.5$) to simulate heterogeneous client distributions. The chosen LLMs include `google/gemma-1.1-2b-it` Team et al. (2024), `tiiuae/Falcon3-3B-Base` Team (2024), and `microsoft/phi-2`Abdin et al. (2023), all of which support fine-tuning via embedding vectors, facilitating evaluation of our method's effectiveness. Additionally, we employ the AdvGLUE dataset Wang et al. (2022) to simulate black-box attacks, evaluating fedEAT's adversarial robustness and performance degradation under such conditions. Detailed experimental settings are provided in the Appendix A.2.

**Baseline:** Given the lack of prior work on adversarial robustness in federated LLMs, we adopt federated adversarial training (FAT) Zizzo et al. (2020)—one of the earliest and most representative adversarial train methods in FL—as our baseline. FAT splits each client's mini-batch into two subsets: one subset of $K$ samples is converted into adversarial examples via FGSM, while the remaining $N - K$ samples remain unperturbed. Gradients from both adversarial and natural samples are then computed and aggregated globally. A detailed explanation is provided in the Appendix A.1.

**Federated LLMs Configuration:** Our federated setup comprises 5 clients and runs for 40 communication rounds. To significantly reduce communication overhead for hundred-billion-parameter

models, we employ the LoRA low-rank adaptation framework illustrated in Figure 2, updating only a small set of low-rank parameters.

**Evaluation Metrics:** We evaluate our method from two perspectives. Robustness evaluation is measured by classification accuracy under adversarial attacks, where higher accuracy indicates stronger resilience to adversarial perturbations. Performance evaluation is measured by classification accuracy on clean test sets, where higher accuracy reflects better baseline task performance.

## 4.2 RESULTS

In this section, we first subject the fedAvg-trained federated LLMs to adversarial attacks of varying intensities to reveal its vulnerability to such perturbations. We then carry out extensive experiments that conclusively demonstrate the effectiveness of our algorithm. For more experimental results, please refer to the Appendix A.3.

**Adversarial Robustness of FedAvg-Trained Federated LLMs under Adversarial Attacks.** Using the `google/gemma-1.1-2b-it` model as a case study, we evaluated the adversarial robustness of a FedAvg-trained federated LLMs across a range of adversarial perturbation strength $\epsilon$ (see Figure 4). The results reveal a dramatic drop in robust accuracy as $\epsilon$ increases, underscoring the acute vulnerability of undefended federated LLMs to adversarial inputs. This finding highlights the critical need for specialized defenses like fedEAT to bolster federated LLMs against adversarial threats.

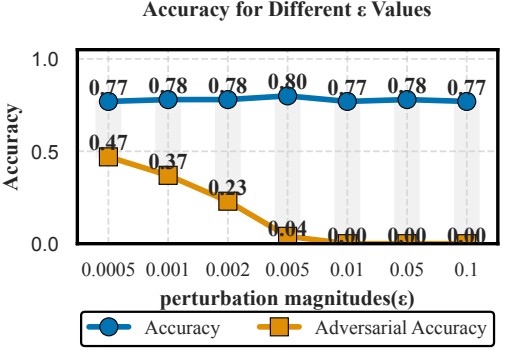

Figure 4: This figure illustrates the robust accuracy under adversarial attacks with varying perturbation magnitudes.

**Medical and Legal Text Classification Results with FGSM.** We evaluated fedAvg, FAT, and fedEAT on four domain-specific datasets—PubMed RCT, Medical Abstracts, SCOTUS, and ECHR—under both clean and adversarial settings. During adversarial training, both FAT and fedEAT by default use FGSM to generate adversarial samples. As shown in Table 2, fedEAT achieves a substantial gain in adversarial accuracy while incurring only a slight change—and in some cases a modest improvement—in clean accuracy. This improvement likely stems from the moderate perturbation strength $\epsilon$, which also enhances generalization. Moreover, fedEAT consistently outperforms FAT across nearly all metrics.

Table 2: Results of Different LLMs across Medical and Legal Text Classification Datasets

| Metric | Method | gemma-1.1-2b-it | | | | Falcon3-3B-Base | | | | phi-2 | | | |
| | | ECHR | MedAbs | PubMed | SCOTUS | ECHR | MedAbs | PubMed | SCOTUS | ECHR | MedAbs | PubMed | SCOTUS |
|---|---|---|---|---|---|---|---|---|---|---|---|---|---|
| | fedavg | 34.5% | 48.3% | 4.1% | 41.4% | 45.7% | 55.4% | 43.9% | 47.2% | 25.2% | 47.5% | 26.4% | 53.3% |
| Adv. Acc (↑) | fedEAT | **46.0%** | **57.6%** | **60.4%** | **54.8%** | **55.2%** | **59.9%** | **70.6%** | **61.9%** | 42.3% | **64.1%** | 67.3% | **57.9%** |
| | FAT | 42.7% | 55.0% | 55.8% | 52.3% | 52.6% | 58.5% | 66.8% | 57.4% | **44.3%** | 64.0% | **69.1%** | 55.9% |
| | fedavg | **51.4%** | 60.9% | **79.5%** | 61.2% | 59.3% | 64.3% | 78.6% | 60.7% | **46.5%** | **69.5%** | **80.3%** | 59.9% |
| Acc (↑) | fedEAT | 50.8% | 61.1% | 77.6% | **61.6%** | 59.3% | **64.4%** | **79.9%** | **65.7%** | 43.4% | 66.3% | 78.8% | **65.2%** |
| | FAT | 50.9% | **63.6%** | 40.3% | 60.9% | 58.7% | 58.8% | 75.1% | 64.5% | 45.7% | 66.1% | 79.0% | 61.4% |

**Note**: Dataset abbreviations: ECHR = European Court of Human Rights, MedAbs = Medical Abstracts, PubMed = PubMed Randomized Controlled Trials

**Medical and Legal Text Classification Results with PGD-3.** When we replace FGSM with PGD-3 for generating adversarial samples in both the adversarial training and adversarial robustness evaluating of fedEAT and FAT, Table 3 reports results that mirror those in Table 2: fedEAT's robustness increases markedly with minimal impact on clean accuracy. These results collectively demonstrate that incorporating adversarial training in the embedding space within federated LLMs can significantly enhance their adversarial robustness while preserving their original performance.

Table 3: Results of Different LLMs across Medical and Legal Text Classification Datasets with PGD-3

| Metric | Method | gemma-1.1-2b-it | | | | Falcon3-3B-Base | | | | phi-2 | | | |
|---|---|---|---|---|---|---|---|---|---|---|---|---|---|
| | | ECHR | MedAbs | PubMed | SCOTUS | ECHR | MedAbs | PubMed | SCOTUS | ECHR | MedAbs | PubMed | SCOTUS |
| Acc (↑) | fedEAT | 48.65% | **61.50%** | 74.88% | 51.16% | 55.33% | **65.75%** | 79.13% | **65.84%** | **46.55%** | 66.00% | **79.63%** | 58.09% |
| | FAT | 45.27% | 58.38% | 71.25% | 49.50% | 52.46% | 65.00% | 78.63% | 64.19% | 43.54% | **68.25%** | 77.75% | **60.13%** |
| | fedavg | **50.18%** | 60.00% | **75.50%** | **60.40%** | **59.72%** | 62.88% | 79.00% | 63.20% | **46.55%** | 66.25% | 79.13% | 56.27% |
| Adv.Acc (↑) | fedEAT | 41.83% | 48.00% | 53.38% | 27.72% | 46.55% | 63.88% | 73.50% | 52.64% | 44.54% | 58.13% | 69.88% | 68.68% |
| | FAT | 38.96% | 47.63% | 42.38% | 25.58% | 43.78% | 55.50% | 71.75% | 50.66% | 41.78% | 53.75% | 66.13% | 58.58% |
| | fedavg | 4.25% | 3.25% | 0.13% | 1.82% | 33.63% | 41.63% | 44.00% | 28.38% | **44.54%** | 46.25% | 21.63% | 48.35% |

**Cross-Domain Text Classification Results.** To demonstrate the scalability of fedEAT, we conducted experiments on four public benchmarks—IMDB, AG News, BBC News, and Reuters—and report the results in Table 4. Across all datasets, fedEAT delivers substantial improvements in adversarial robustness while exerting only a negligible impact on clean accuracy, confirming its effectiveness in diverse real-world scenarios.

Table 4: Results of Different LLMs across Other Text Classification Datasets

| Metric | Method | gemma-1.1-2b-it | | | | Falcon3-3B-Base | | | | phi-2 | | | |
|---|---|---|---|---|---|---|---|---|---|---|---|---|---|
| | | imdb | agnews | bbcnews | reuters | imdb | agnews | bbcnews | reuters | imdb | agnews | bbcnews | reuters |
| Adv.Acc (↑) | fedavg | 19.8% | 0.2% | 15.8% | 1.7% | 44.6% | 24.9% | 50.3% | 37.3% | 16.1% | 15.3% | 55.4% | 17.3% |
| | fedEAT | **98.5%** | **99.9%** | **79.3%** | 61.5% | **78.3%** | **97.5%** | 79.8% | **91.7%** | **84.0%** | **69.0%** | **80.9%** | **51.5%** |
| | FAT | 94.3% | 95.6% | 72.3% | **67.9%** | 77.1% | 83.0% | **81.2%** | 87.7% | 63.9% | 56.8% | 69.3% | 45.1% |
| Acc (↑) | fedavg | 94.1% | **90.8%** | 97.8% | 95.3% | **94.1%** | **92.2%** | 98.4% | 94.7% | 93.6% | 90.4% | **98.8%** | 95.3% |
| | fedEAT | **94.2%** | 90.6% | **97.9%** | **95.8%** | 92.5% | 91.0% | 96.7% | **95.0%** | **94.4%** | **91.1%** | 97.8% | 94.8% |
| | FAT | 93.8% | 84.0% | 97.7% | 94.9% | 90.3% | 86.6% | 95.9% | 93.6% | 92.9% | 90.1% | 97.5% | **96.1%** |

**Adversarial Robustness of FedEAT Under Black-Box Attacks.** To validate the effectiveness of fedEAT under black-box attacks and to assess the performance and adversarial robustness of federated LLMs on natural language understanding tasks, we trained `ZEPHYR-7B`Tunstall et al. (2023) via fedavg and fedEAT; on the SST-2, QQP, MNLI, and QNLI sub-tasks, we conducted comprehensive evaluations using the clean dataset GLUE Wang et al. (2018) and the adversarial dataset advGLUE Wang et al. (2022).

Task performance is measured by classification accuracy, while adversarial robustness is quantified by the Attack Success Rate (ASR), defined as $\text{ASR} = A_m/B_c$, where $B_c$ is the number of correctly classified samples under clean dataset, and $A_m$ is the subset of those samples misclassified after adversarial perturbation. A lower ASR indicates stronger resilience to noise or malicious attacks.

Table 5: Robustness under black-box attacks

| Metric | SST2 | | QQP | | MNLI | | QNLI | |
|---|---|---|---|---|---|---|---|---|
| | fedAvg | fedEAT | fedAvg | fedEAT | fedAvg | fedEAT | fedAvg | fedEAT |
| ASR (↓) | 0.213 | **0.226** | 0.150 | **0.060** | 0.145 | **0.121** | 0.196 | **0.168** |
| Accuracy (↑) | **81.6%** | 77.1% | **70.2%** | 68.8% | **57.0%** | 55.0% | **82.9%** | 81.5% |

As shown in Table 5, fedEAT incurs only a negligible drop in clean accuracy compared to fedAvg, yet it substantially reduces ASR across all tasks, confirming its ability to enhance robustness without sacrificing baseline performance.

**Effectiveness of FedEAT under Varying Attack Strengths.** To assess fedEAT's effectiveness under varying attack strengths, we fix the perturbation strength $\epsilon$ during adversarial sample generation and evaluate the federated LLMs' adversarial robustness using 1-step, 10-step, and 20-step PGD iterations. As shown in Table 6, fedEAT consistently achieves the highest adversarial accuracy in most settings while maintaining strong clean accuracy. These results demonstrate that fedEAT effectively balances model performance and adversarial resilience across a range of attack strengths.

**Ablation Study.** We conduct ablation studies on the Google Gemma-1.1-2b-it model using the IMDB dataset, focusing on two key parameters: the perturbation strength $\epsilon$ (ranging from 0.05 to 5.0) and the adversarial loss weight $\lambda$ (ranging from 0.0 to 1.0), as shown in Figure 5. Results show that when $\epsilon$ is

Table 6: Results of Federated gemma-1.1-2b-it Model Across Different PGD Steps

| Dataset | Method | Acc(↑) | Adv.Acc (↑) | | |
|---|---|---|---|---|---|
| | | | PGD Steps | | |
| | | | 1 | 10 | 20 |
| agnews | fedEAT | **91.67%** | **76.38%** | **75.50%** | **78.00%** |
| | FAT | 89.04% | 71.00% | 66.50% | 73.50% |
| | fedavg | 90.29% | 19.00% | 16.50% | 17.37% |
| bbcnews | fedEAT | 98.23% | 92.24% | **93.67%** | **93.67%** |
| | FAT | 98.23% | **92.65%** | 92.45% | 93.27% |
| | fedavg | **98.50%** | 86.12% | 82.86% | 81.02% |
| imdb | fedEAT | 93.50% | 64.12% | **70.0%** | **73.12%** |
| | FAT | 92.08% | **70.00%** | 68.12% | 69.38% |
| | fedavg | **93.96%** | 55.25% | 46.75% | 51.88% |
| reuters | fedEAT | 95.63% | **86.57%** | **87.70%** | **88.51%** |
| | FAT | 95.41% | **86.57%** | 87.06% | 86.25% |
| | fedavg | **95.69%** | 71.20% | 57.28% | 61.33% |

between 0.05 and 0.5, the model maintains a natural accuracy above 90% while achieving significant improvements in adversarial robustness. However, when $\epsilon$ exceeds 1.0, although robustness continues to improve, natural accuracy drops sharply to around 50%. For $\lambda$, a range of 0.1–0.7 achieves a good balance between robustness and natural accuracy. Beyond this range, the model either struggles with adversarial inputs or suffers a significant drop in clean-data accuracy.

Overall, the ablation results indicate that fedEAT exhibits strong robustness to parameter choices: across a relatively broad range of $\epsilon = 0.05$–$0.5$ and $\lambda = 0.1$–$0.7$, the model consistently achieves both high clean accuracy and strong adversarial robustness. This makes fedEAT adaptable to a wide variety of application scenarios. Based on these findings, we adopt $\epsilon = 0.1$ and $\lambda = 0.5$ as the default settings in our main experiments, striking a favorable balance between robustness and performance.

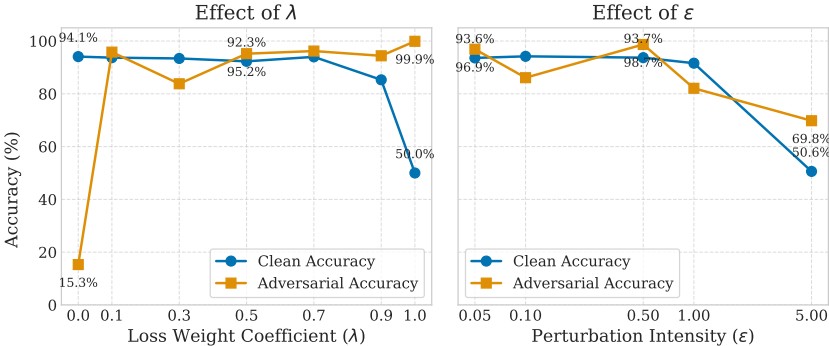

Figure 5: Ablation study results of key parameters in the fedEAT method. The left figure shows the effect of loss weight coefficient $\lambda$ on model performance. The right figure illustrates the impact of perturbation intensity parameter $\epsilon$ on model performance.

## 5 CONCLUSION

Federated large language models (LLMs) are poised to play a pivotal role in the future of AI. To address their vulnerability to adversarial attacks, we introduce fedEAT, which combines embedding-space adversarial training with LoRA low-rank adaptation. Our experiments show that fedEAT significantly bolsters adversarial robustness across multiple domains and tasks with minimal impact on original performance, and ablation studies confirm its efficacy and cross-domain scalability. This work provides a systematic approach for enhancing the adversarial resilience of federated LLMs.

## 6 Reproducibility Statement

To ensure the reproducibility of our experiments and results, we provide the complete codebase, all reported results, and detailed usage instructions at the following anonymous link: `https://anonymous.4open.science/r/federatedLLM-robust-F01C/README.md`. Our implementations, including the proposed method and all baselines, are available; the hyperparameter tuning, training, and evaluation pipelines are included; the hyperparameters used are fully documented; the datasets and their splits are provided; and within a fixed software environment, most results are bitwise reproducible.

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

Table 7: Comparison with Related Work to FedEAT

| Study | Setting | Core Method | Difference from FedEAT |
|---|---|---|---|
| Li et al. Xhonneux et al. (2024) (2024) | Centralized LLM | Embedding-space adversarial training; *toward/away/utility* joint loss | Targets only centralized generation tasks; limited scalability; requires manual harmful-prompt dataset and extra utility data. |
| Wang et al. Mazeika et al. (2024) (2024) | Centralized LLM | Dynamic red-team prompt pool; *toward/away/utility* joint loss | Inherits Li et al.'s limitations; generates adversarial samples in discrete text space—high computational cost. |
| Zizzo et al. Zizzo et al. (2020) (2020) | Federated Learning | Split client batch: PGD adversarial vs. natural training | Designed for small CNNs on image tasks; high compute and communication cost for large LLMs. |
| Zhong et al. Zhong et al. (2023) (2023) | Centralized LLM | Adversarial training as mixed-strategy game with sampling | Validated only on BERT-scale models; may incur prohibitive overhead on larger LLMs. |
| Zhang et al. Zhang et al. (2023) (2023) | Federated Learning | Decision-boundary federated adversarial training (DBFAT) | Tailored to small models and image tasks; faces similar cost issues on LLMs. |
| Bhagoji et al. Bhagoji et al. (2019)] (2019) | Federated Learning | Theoretical adversarial analysis | Purely theoretical; no distributed adversarial training algorithm. |

## A  RELATED WORKS

Table 7 summarizes several works that are closely related to ours.

### A.1  BASELINE ALGORITHM: FAT

As adversarial robustness of federated LLMs remains underexplored, we adopt the classic Federated Adversarial Training (FAT) Zizzo et al. (2020) as our baseline. In FAT, each client splits its local mini-batch into two subsets—$K/N = 0.5$ examples for adversarial training and $N - K$ examples for natural training. Adversarial samples are generated via PGD-3 in the embedding space (see Algorithm 2, Difference 1), while natural samples are left unperturbed. The losses from both subsets are summed directly without an additional trade-off coefficient $\lambda$ (Difference 2). Clients perform one gradient update per round, and the server aggregates all updates by simple averaging to form the next global model. Unlike FedEAT, FAT does not compute a per-sample weighted combination of standard and adversarial losses.

---

**Algorithm 2** FAT: Federated Adversarial Training (Embedded-space Version)

---

**Require:** Global model parameters $\mathbf{w}^{(0)}$, local datasets $\{D_k\}_{k=1}^{n}$, communication rounds $T$, local epochs $E$, learning rate $\eta$, perturbation step $\alpha$, norm bound $\epsilon$, adversarial ratio $K/N$, iteration rounds $I$

**Ensure:** Global model parameters $\mathbf{w}^{(T)}$

1: **for** $t = 0$ to $T - 1$ **do**
2:     Server broadcasts $\mathbf{w}^{(t)}$ to all clients
3:     **for** each client $k = 1, \ldots, n$ **in parallel do**
4:         $\theta_k \leftarrow \mathbf{w}^{(t)}$ {Initialize local model}
5:         **for** $e = 1$ to $E$ **do**
6:             Sample mini-batch $\mathcal{B}$ of size $N$ from $D_k$
7:             Split $\mathcal{B}$ into $\mathcal{B}_{\text{adv}}$ ($K$ samples) and $\mathcal{B}_{\text{nat}}$ ($N - K$ samples) {Difference 1: batch split}
8:             $L \leftarrow 0$
9:             **for** each $(x, y) \in \mathcal{B}_{\text{adv}}$ **do**
10:                $\mathbf{z} \leftarrow \text{Embed}(x)$ {Generate embedding vector}
11:                $\delta^0 \leftarrow \mathbf{0}$ {Initialize perturbation vector}
12:                **for** $i = 0$ to $I$ **do**
13:                   $\delta^{i+1} \leftarrow \Pi_{\mathcal{B}(0, \epsilon)}\big(\delta^i + \alpha \cdot \text{sign}(\nabla_{\mathbf{z}} \mathcal{L}(f_\theta(\mathbf{z} + \delta^i), y))\big)$
14:                **end for**
15:                $\mathbf{z}_{\text{adv}} \leftarrow \mathbf{z} + \delta^3$ {Generate adversarial sample}
16:                $L \leftarrow L + \mathcal{L}(f_\theta(\mathbf{z}_{\text{adv}}), y)$
17:             **end for**
18:             **for** each $(x, y) \in \mathcal{B}_{\text{nat}}$ **do**
19:                $\mathbf{z} \leftarrow \text{Embed}(x)$ {Generate embedding vector}
20:                $L \leftarrow L + \mathcal{L}(f_\theta(\mathbf{z}), y)$
21:             **end for**
22:             $\theta_k \leftarrow \theta_k - \eta \nabla_{\theta_k} L$ {Difference 2: no trade-off coefficient}
23:         **end for**
24:         Client $k$ returns $\theta_k$ to the server
25:     **end for**
26:     Server aggregation: $\mathbf{w}^{(t+1)} \leftarrow \frac{1}{n} \sum_{k=1}^{n} \theta_k$ {Simple averaging}
27: **end for**
28: **return** $\mathbf{w}^{(T)}$

---

## A.2 EXPERIMENTAL SETUP

This appendix details the configuration parameters used in our federated LLMs experiments.

### A.2.1 HARDWARE

All experiments were conducted on NVIDIA A6000 GPUs with 48 GB of VRAM, requiring at least 720 GPU hours in total.

### A.2.2 MODEL CONFIGURATION

In this study, we fine-tuned three LLMs using LoRA (Low-Rank Adaptation) technique. We defined the task type as sequence classification, which aligns with the nature of our target tasks. To effectively reduce the parameter count while maintaining model performance, we set the LoRA rank ($r$) to 8, which demonstrated an excellent balance between performance and efficiency in our preliminary experiments. The LoRA scaling factor (alpha) was set to 32.

### A.2.3 DATASET CONFIGURATION

Considering the parameter scale of large language models and computational resource constraints, we limited each dataset to a maximum of 2,000 samples. This sample size ensures the model can learn sufficient data patterns while maintaining training efficiency. To standardize input formats and optimize memory usage, we restricted the maximum sequence length to 256 tokens, which

is sufficient to capture essential semantic information for most classification tasks while avoiding unnecessary computational overhead.

In our experiments, each dataset was first split into training and test sets by stratified sampling with a 60%/40% ratio to maintain consistent label distributions. For the training data, we set $\alpha = 0.5$ and employed a Dirichlet distribution to generate class proportions across 5 clients, distributing samples per class randomly according to these proportions. This approach preserves the representativeness of the test set while flexibly simulating a moderate non-IID data distribution.

### A.2.4 FEDERATED LEARNING CONFIGURATION

Within our federated LLMs framework, we designed a system comprising 5 clients to simulate data silos in real-world distributed environments. To comprehensively evaluate model performance, we employed a client participation ratio of 1.0, meaning all clients participated in model updates during each communication round. The number of communication rounds was set to 40, which our preliminary experiments indicated was sufficient for model convergence. After each communication round, clients performed only 1 local training epoch.

### A.2.5 TRAINING HYPERPARAMETERS

To accommodate federated LLMs training under limited VRAM conditions, we set the batch size to 8. This relatively small value helps reduce memory consumption while maintaining reasonable gradient estimation accuracy. The learning rate was set to $2 \times 10^{-4}$, and optimization was performed using the AdamW optimizer. To ensure reproducibility of experimental results, we used 42 as the random seed across all experiments to control randomness in initialization processes and data partitioning. These hyperparameter choices carefully consider the characteristics of LLM fine-tuning and the constraints of federated learning environments, aiming to achieve an optimal balance between performance and efficiency.

### A.2.6 EXPERIMENTAL SETUP FOR BLACK-BOX ATTACKS ON FEDEAT

To evaluate fedEAT's adversarial robustness under black-box attacks, we selected the GLUE test sets (SST-2, QQP, MNLI, and QNLI) from Wang et al. Wang et al. (2018) as clean data and simulated black-box attacks using the advGLUE adversarial samples Wang et al. (2022). We further leveraged the *vicgalle/alpaca-gpt4* dataset to generate additional adversarial examples via a proxy model's subtle semantic or syntactic perturbations to strengthen adversarial training. Two model variants—ZEPHYR-7B (7B parameters)—were trained under both FedAvg and fedEAT aggregation schemes, using 5 clients, 1 local epoch per round, a learning rate of $2 \times 10^{-4}$, and a batch size of 8; within fedEAT, we applied PGD-based adversarial fine-tuning with $\epsilon = 0.01$. We assessed task utility by accuracy on the clean test sets and quantified robustness via the Attack Success Rate (ASR):

$$\text{ASR} = \frac{A_m}{B_c},$$

where $B_c$ is the number of correctly classified clean samples and $A_m$ is the number of those samples misclassified under adversarial perturbations. A lower ASR indicates stronger resistance to black-box attacks. Example clean/adversarial pairs are shown in Table 8.

### A.3 RESULTS

### A.3.1 ADVERSARIAL ATTACKS OF VARYING $\epsilon$ ON FEDAVG-TRAINED FEDERATED LLMS

We subjected the federated LLMs—trained via the fedAvg algorithm—to adversarial perturbations of increasing magnitude. Specifically, we varied the attack budget $\epsilon$ over the set

$$\epsilon \in \{0.0005, 0.001, 0.002, 0.005, 0.01, 0.05, 0.1\}.$$

Figure 6 demonstrate that the model's accuracy remains essentially unchanged under clean (unperturbed) test data. However, as $\epsilon$ increases, the model's adversarial accuracy steadily declines and ultimately collapses to zero. This dramatic degradation highlights the extreme vulnerability of federated LLMs to adversarial attacks, and thereby motivates our development of algorithms to enhance their adversarial robustness.

Table 8: Comparison of clean (Original) and Adversarial (Modified) Dataset Examples

| INDEX | TASK | TYPE | PROMPT (QUESTION/SENTENCE) | LABEL |
|---|---|---|---|---|
| 1 | QQP | ORIGINAL | QUESTION1: CAN EATING ONLY FRUIT FOR DINNER LEAD TO WEIGHT LOSS ? QUESTION2: DOES EATING DINNER EARLIER IN THE EVENING HELP WITH WEIGHT LOSS ? | 0 (NO) |
| | | MODIFIED | QUESTION1: CAN EATING PRECISELY ADVANCE FOR DINNER LEAD TO WEIGHT LOSS ? QUESTION2: DOES EATING DINNER EARLIER IN THE EVENING HELP WITH WEIGHT LOSS ? | 0 (NO) |
| 187 | SST2 | ORIGINAL | SENTENCE: A SUBTLE AND WELL-CRAFTED ( FOR THE MOST PART ) CHILLER | 1 (POSITIVE) |
| | | MODIFIED | SENTENCE: A SUBTLE AND WELL-CRAFTED ( FOR THE MOST PART ) CHILLER AND FALSE IS NOT TRUE . | 1 (POSITIVE) |
| 300 | QNLI | ORIGINAL | QUESTION: WHAT KIND OF NEEDLEWORK WAS USED IN THE CREATION OF THE BAYEUX TAPESTRY ? ANSWER: IT WAS COMMISSIONED BY ODO , THE BISHOP OF BAYEUX AND FIRST EARL OF KENT , EMPLOYING NATIVES FROM KENT WHO WERE LEARNED IN THE NORDIC TRADITIONS IMPORTED IN THE PREVIOUS HALF CENTURY BY THE DANISH VIKINGS . | 1 (YES) |
| | | MODIFIED | QUESTION: WHAT KIND OF NEEDLEWORK WAS USED IN THE CREATION OF THE BAYEUX TAPESTRY ? ANSWER: MARITIME WAS SEWING PLATES ODO , THE BISHOP OF BAYEUX AND FIRST EARL OF KENT , EMPLOYING NATIVES FROM KENT WHO WERE LEARNED IN THE NORDIC TRADITIONS IMPORTED IN THE PREVIOUS HALF CENTURY BY THE DANISH VIKINGS . | 1 (YES) |

Figure 6: This figure shows the standard and adversarial accuracy of a fedavg-trained federated LLMs, measured every 5 training rounds, under adversarial attacks of varying perturbation strengths ($\epsilon$). As $\epsilon$ increases, the model's robust accuracy consistently declines—highlighting its vulnerability to stronger adversarial perturbations.

### A.3.2 CENTRALIZED VS. FEDERATED LLMS: PERFORMANCE AND ADVERSARIAL ROBUSTNESS

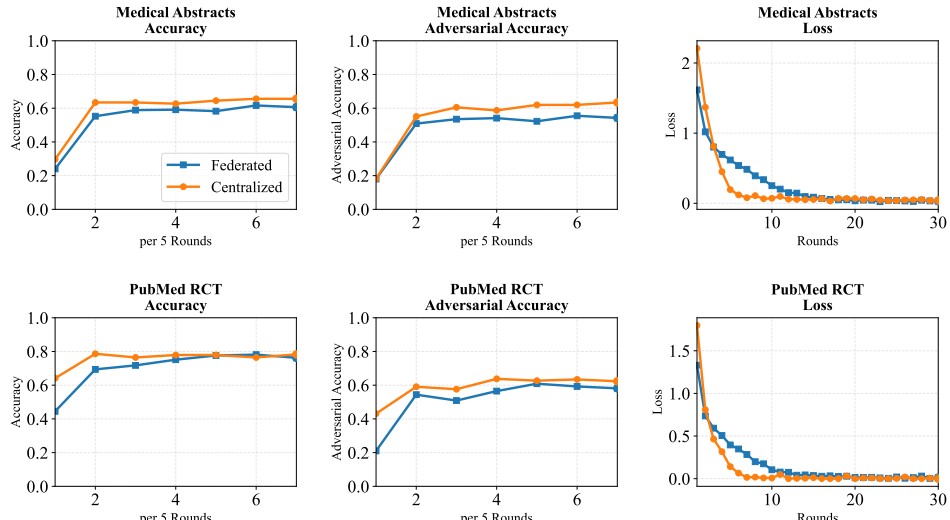

Figure 7: This figure illustrates the accuracy, adversarial accuracy, and training loss of the `gemma-1.1-2b-it` on the Medical Abstracts and PubMed RCT datasets. Accuracy and adversarial accuracy were measured at every 5 communication rounds.

To quantify the gap between federated and centralized LLMs, we applied the fedEAT to both federated LLMs (with 3 clients) and its centralized counterpart. As shown in Figure 7, the federated LLMs consistently achieves slightly lower clean accuracy and robust accuracy than the centralized LLM, and its convergence is noticeably slower. Nonetheless, the performance drop is modest, underscoring the practical viability of federated LLMs in privacy-sensitive settings.

### A.3.3 COMPLETE RESULTS OF ABLATION STUDIES

Table 9: Ablation results for perturbation magnitude $\epsilon$

| $\epsilon$ | Acc (↑) | | Adv. Acc (↑) | |
|---|---|---|---|---|
| | Max | Avg$_{\text{last-5}}$ | Max | Avg$_{\text{last-5}}$ |
| 0.05 | 94.9% | 93.6% | 99.5% | 96.9% |
| 0.10 | 95.3% | 94.2% | 99.0% | 86.1% |
| 0.50 | 94.6% | 93.7% | 99.5% | 98.7% |
| 1.00 | 94.0% | 91.6% | 98.4% | 82.1% |
| 5.00 | 52.1% | 50.6% | 100.0% | 69.8% |

Tables 9 and 10 present the complete results of ablation experiments for two key parameters in the fedEAT method. For each table, the first two columns show the maximum accuracy achieved across all communication rounds and the average accuracy of the last five communication rounds, while the last two columns display the corresponding adversarial accuracy metrics. These detailed data provide a comprehensive perspective that helps understand how these parameters influence model performance and robustness.

For the perturbation magnitude parameter $\epsilon$ (Table 9), the data demonstrates that within a smaller range (0.05-0.50), the model maintains both high natural accuracy and strong adversarial robustness. As $\epsilon$ increases to 5.00, natural accuracy drops significantly, confirming that excessive perturbation damages the model's fundamental performance. Notably, when $\epsilon = 1.00$, the average adversarial accuracy shows a relative decline, suggesting a non-linear relationship where overly large perturbations may lead to training instability.

Table 10: Ablation results for loss weight coefficient $\lambda$

| $\lambda$ | Acc ($\uparrow$) | | Adv. Acc ($\uparrow$) | |
|---|---|---|---|---|
| | Max | $\text{Avg}_{\text{last-5}}$ | Max | $\text{Avg}_{\text{last-5}}$ |
| 0.0 | 94.3% | 94.1% | 18.3% | 15.3% |
| 0.1 | 94.3% | 93.7% | 99.5% | 95.8% |
| 0.3 | 95.4% | 93.4% | 99.5% | 83.8% |
| 0.5 | 93.4% | 92.3% | 99.9% | 95.2% |
| 0.7 | 94.8% | 94.0% | 98.8% | 96.2% |
| 0.9 | 94.8% | 85.3% | 99.1% | 94.4% |
| 1.0 | 54.8% | 50.0% | 100.0% | 99.9% |

For the loss weight coefficient $\lambda$ (Table 10), the data clearly illustrates its crucial role. When $\lambda = 0.0$ (i.e., no adversarial training), the model almost completely fails against adversarial examples. Introducing even minimal adversarial training ($\lambda = 0.1$) significantly improves adversarial accuracy while maintaining high natural accuracy. Within the range of $\lambda$ values from 0.1 to 0.7, the model exhibits stable natural and adversarial accuracy, demonstrating good parameter adaptability. However, when $\lambda$ approaches or equals 1.0, excessive weighting of adversarial loss causes a dramatic decline in the model's performance on natural samples.

These detailed results not only validate our observations in the main analysis but further indicate that the fedEAT method offers considerable flexibility in parameter selection, achieving good balance across multiple configurations. This characteristic has important implications for applying fedEAT to tasks of varying complexity and models of different scales, enhancing the method's practicality and versatility.

### A.4 LIMITATIONS

Although fedEAT demonstrates strong empirical performance and adversarial robustness, it has the following limitations that warrant future investigation:

- **Dependence on embedding-vector input.** Our approach requires models to accept and process external embedding vectors. It cannot be directly applied to closed-source LLMs that only expose discrete-text APIs. Future work will explore general adversarial training strategies compatible with all major LLM interfaces.

- **Task and domain scope.** We focus exclusively on text classification tasks using public benchmarks. fedEAT's effectiveness on text generation, question answering, and other downstream or cross-domain scenarios remains unverified. We plan to extend our evaluation to a broader set of language tasks and assess cross-domain adversarial generalization.

- **Lack of theoretical convergence and robustness guarantees.** This work provides empirical validation of fedEAT but does not offer rigorous theoretical analysis of convergence or provable robustness for distributed embedding-space adversarial training and LoRA aggregation. Future research will incorporate distributed optimization theory to establish formal guarantees.

### A.5 BROADER IMPACTS

In privacy-sensitive domains such as healthcare, finance, and law, fedEAT enhances the adversarial robustness of federated LLMs, reducing the risk of misdiagnoses, faulty risk assessments, and retrieval errors caused by malicious or inadvertent inputs, thereby improving system security and reliability. Its low-rank adapter sharing mechanism further protects data sovereignty. On the downside, adversaries could co-opt similar techniques to fortify their own models against detection, potentially escalating the arms race between attack and defense. Moreover, the extra computation required for large-scale adversarial training increases energy consumption and carbon footprint.

## A.6    THE USE OF LARGE LANGUAGE MODELS (LLMS)

In this work, Large Language Models (LLMs) were used solely as a general-purpose writing assistance tool to help polish the text for clarity, grammar, and readability. All content generated by the LLMs was carefully reviewed by the authors, and only the parts considered accurate and appropriate were incorporated. The research ideation, methodology, experiments, and results were entirely developed and conducted by the authors. We take full responsibility for the content of the paper, including any portions that were influenced by LLM-assisted text editing.

