# OpenReview forum: "FedEAT: A Robustness Optimization Framework for Federated LLMs"
_ICLR.cc/2026/Conference — Submitted to ICLR 2026_

### Official Review · Reviewer_LMBq · 2025-10-24

**Soundness:** 3
**Presentation:** 3
**Contribution:** 2
**Rating:** 4
**Confidence:** 3

**Summary:**

This paper introduces FedEAT, a novel federated embedding-space adversarial training method aimed at improving the robustness of federated large language models (LLMs). The approach combines embedding-space adversarial perturbations, a regularization-based robustness–accuracy trade-off, and LoRA-based parameter-efficient fine-tuning to address the computational and communication challenges of adversarial training in federated environments.

Empirical results across multiple domains—including healthcare, finance, law, and general text classification—demonstrate that FedEAT achieves substantial gains in adversarial robustness with minimal degradation in clean accuracy. The experiments are extensive, covering various LLM backbones, adversarial attack types (FGSM, PGD, and black-box), and ablation studies on key hyperparameters. The work is clearly written, methodologically sound, and provides reproducible code and documentation.

**Strengths:**

- The paper tackles an important and underexplored problem: adversarial robustness in federated large language models.

- The embedding-space adversarial training idea is a logical and computationally efficient adaptation of prior adversarial training strategies to the federated context.

- Empirical results are broad, covering several model architectures and domains.

- The algorithm is clearly formulated, and the code availability supports reproducibility.

**Weaknesses:**

1. Limited Novelty
- The proposed method is a straightforward combination of known techniques, primarily adversarial training in embedding space (Xhonneux et al., 2024) and LoRA-based federated fine-tuning.
- There is no fundamentally new optimization principle or theoretical advancement; the contribution is mainly empirical.

2. Lack of Theoretical Foundation
- The work lacks formal analysis of robustness guarantees, convergence properties, or communication complexity.

- Claims of efficiency and scalability remain qualitative, with no quantitative analysis of computational overhead or gradient stability.

3. Experimental Design Limitations

- Experiments use small-scale datasets (≤2k samples per client) and medium-sized models (2B–3B parameters), which limits the generalizability of the conclusions to realistic trillion-parameter federated LLM settings.

- Hyperparameter choices (λ, ε) are manually tuned without systematic justification or sensitivity analysis.

- Statistical significance and variability of results are not reported.

4. Presentation and Discussion Gaps

- The paper overstates its conceptual novelty and underexplores broader implications such as privacy–robustness trade-offs.

- Discussion of potential failure cases or limitations is minimal.

- The narrative could be more concise, with stronger emphasis on insights rather than experimental enumeration.

**Questions:**

1. Quantitative Efficiency Analysis
   The paper claims that FedEAT reduces computational and communication overhead compared to standard adversarial training. Could the authors quantify the actual reduction in computation time, GPU memory usage, and communication cost relative to FAT or FedAvg? Concrete benchmarks (e.g., FLOPs, training hours, transmitted parameters) would clarify the claimed efficiency.

2. Scalability to Larger Models
   Experiments are conducted with 2B–3B parameter models and small datasets. How would FedEAT scale to trillion-parameter LLMs or production-scale FL environments where clients may have heterogeneous hardware? Is the embedding-space adversarial generation still feasible under such constraints?

3. Trade-off Parameter (λ) Selection
   The paper uses λ = 0.5 as the default but provides limited justification. Could the authors explain how λ was chosen, whether automatic tuning or adaptive scheduling was explored, and how sensitive the performance is to this hyperparameter?

4. Adversarial Perturbation Bound (ϵ)
   Since ϵ directly controls the perturbation magnitude, can the authors provide quantitative guidelines or heuristics for selecting it across datasets? Is there a correlation between data domain characteristics (e.g., vocabulary diversity, text length) and optimal ϵ values?

6. Robustness Evaluation Scope
   The evaluation mainly considers FGSM, PGD, and black-box attacks. Have the authors tested prompt-level or instruction-level adversarial attacks, which are more representative of real-world LLM threats? This would strengthen the claim of “robust federated LLMs.”

7. Communication Strategy under LoRA
   The paper integrates LoRA to reduce communication overhead, but the communication protocol is not deeply discussed. Could the authors provide details on parameter synchronization frequency, compression ratio, or bandwidth assumptions?

---

### Official Review · Reviewer_jocL · 2025-10-28

**Soundness:** 2
**Presentation:** 2
**Contribution:** 2
**Rating:** 4
**Confidence:** 4

**Summary:**

This paper proposes FedEAT, a framework to address the vulnerability of federated LLMs to adversarial attacks. The method performs adversarial training by generating perturbations directly in the continuous embedding space of the client LLMs. This approach is combined with the LoRA training strategy to reduce communication overhead. A regularization term is also included in the loss function to balance the model's performance on clean inputs against its robustness to adversarial examples.

**Strengths:**

The proposed method is clearly illustrated through a well-structured flow diagram, and the experimental evaluation covers a broad range of datasets. The ablation experiments are also relatively comprehensive and provide useful insights.

**Weaknesses:**

1. The paper claims to address two specific challenges—compute and memory efficiency, and distributed robustness under communication bottlenecks. However, the proposed method does not provide a detailed explanation of how each of these challenges is explicitly handled. In addition, the experiments do not clearly demonstrate how the proposed approach mitigates these issues. It remains unclear whether the main novelty lies solely in the introduction of a regularization term, as the adversarial sample generation process appears to follow existing methods.

2. Regarding the memory efficiency challenge, the introduction of adversarial sample generation seems to increase the memory footprint on client devices rather than reduce it. Please clarify how this process aligns with the claimed improvement in memory efficiency.

3. The notation used in Figure 4 is not well explained, and its meaning is inconsistent with symbols used elsewhere in the paper. Please ensure consistent notation and provide a legend or clarification in the caption.

4. The comparative experiments include relatively few baselines, and the most recent one dates back to 2020. It is strongly recommended to compare against more recent methods to better highlight the competitiveness of the proposed approach.

5. As correctly noted in Section A.4, the paper would benefit from including theoretical guarantees on convergence and robustness to strengthen its technical soundness.

6. In Algorithm 2, it is unclear why no additional trade-off coefficient is introduced, given that the training process involves a mixture of natural and adversarial samples.

**Questions:**

Please refer to above

---

### Official Review · Reviewer_mhG2 · 2025-10-30

**Soundness:** 2
**Presentation:** 3
**Contribution:** 2
**Rating:** 2
**Confidence:** 4

**Summary:**

The authors propose a method called FedEAT designed to enhance adversarial robustness in federated learning models. This work  address adversarial robustness in the context of LLM fine-tuning within federated settings. The method is conceptually straightforward, representing a natural extension of existing robustness techniques to the federated learning domain. While the technical approach is relatively simple, the paper does provide experimental analysis of the method's behavior across multiple models and datasets. However, the empirical evidence falls significantly short of supporting the authors'  main claims. Further weaknesses are detailed below

**Strengths:**

1) The paper addresses adversarial robustness at inference time for federated LLMs in federated setting environments where computational and communication limitations prohibit standard multi-step, full-parameter adversarial training approaches.
2) The evaluation provides broad empirical coverage, spanning multiple model architectures, diverse task domains, and various attack types ie: black box/white box
3) The proposed method is both conceptually straightforward and practically implementable.

**Weaknesses:**

The way the experiment is conducted is not completely correct  because of several reasons:
-   The experiments run for just one epoch, which is not really federated in the truest sense, especially if the non-IID setup is created through simple sample distributions across clients. The experiments then just become very similar to  the centralized setting. Since the authors dont even have one ablation of multi epoch training the problem I am not sure if any conclusions can be made.
-  Second, the non-IID setup is not realistic and does not address the true challenges inherent in federated learning settings. There should be feature skew or task skew to simulate true non-IID conditions. For instance, one approach could be to compute embeddings for all samples and use optimal transport to assign batches to clients with a target  Earth Mover Distance (EMD)  band—something along these lines would help to truly simulate heterogeneity in clients.
-   Is a sufficient gap in the adversarial examples on each client generated, or is it lost through mere aggregation?
-   How do different delta or epsilon values per client affect the federated setting, and what challenges are introduced?
- It’s unclear whether adversarial gains are uniform across clients or dominated by a few. The paper should report per-client clean/robust accuracy, gradient conflict metrics, and whether averaging erodes client-specific robustness (especially under heterogeneous data -- which again comes back to the way non-IID is implemented).

I am unconvinced by the experiment  in this paper.

**Questions:**

1) Does aggregation preserve or diminish the diversity of adversarial examples generated on each client? Is there sufficient variation in the adversarial perturbations across clients, or does the aggregation process homogenize them?

2) How do heterogeneous privacy budgets (different δ or ε values per client) affect federated training dynamics? What specific challenges does this heterogeneity introduce for convergence and robustness?

3)Can the authors provide per-client performance metrics (clean accuracy, robust accuracy) and analyze gradient conflicts during aggregation? Does the averaging process erode robustness gains achieved on individual clients, especially under more realistic non-IID conditions?

---

### Meta-Review · Area_Chair_NTDq · 2026-01-21

**Summary:**

The paper proposes a method for improving adversarial robustness in federated LLM fine-tuning by performing adversarial training directly in the embedding space on the clients, using a regularization term to balance robustness vs. clean accuracy, and using LoRA fine-tuning to reduce communication overhead. Reviewers found several issues that were not addressed in a rebuttal: they questioned the novelty of the approach, found the experiments lacked sufficient scope (e.g. single epoch experiments, no addressing of heterogeneity, missing baselines)), and a lack of theoretical guarantees.

**Reviewer Concerns:**

No rebuttal was given, all concerns remain unaddressed: scope of the experiments, novelty of the approach, missing baselines, lack of theory, etc.

**Reviewer Scores:**

NA

---

### Decision · Program_Chairs · 2026-01-26

Reject